# Length-Dependent Electronic Transport Properties of the ZnO Nanorod

**DOI:** 10.3390/mi10010026

**Published:** 2018-12-31

**Authors:** Baorui Huang, Fuchun Zhang, Yanning Yang, Zhiyong Zhang

**Affiliations:** 1School of Information Science Technology, Northwest University, Xi’an 710127, China; ydhbr@yau.edu.cn; 2School of Physics and Electronic Information, Yan’an University, Yan’an 716000, China; ydzfc@yau.edu.cn (F.Z.); yadxyyn@163.com (Y.Y.)

**Keywords:** zinc oxide (ZnO) nanorod, transmission spectrum, transport properties, molecular-projected self-consistent Hamiltonian (MPSH), current–voltage (I–V) curves

## Abstract

The two-probe device of nanorod-coupled gold electrodes is constructed based on the triangular zinc oxide (ZnO) nanorod. The length-dependent electronic transport properties of the ZnO nanorod was studied by density functional theory (DFT) with the non-equilibrium Green’s function (NEGF). Our results show that the current of devices decreases with increasing length of the ZnO nanorod at the same bias voltage. Metal-like behavior for the short nanorod was observed under small bias voltage due to the interface state between gold and the ZnO nanorod. However, the influence of the interface on the device was negligible under the condition that the length of the ZnO nanorod increases. Moreover, the rectification behavior was observed for the longer ZnO nanorod, which was analyzed from the transmission spectra and molecular-projected self-consistent Hamiltonian (MPSH) states. Our results indicate that the ZnO nanorod would have potential applications in electronic-integrated devices.

## 1. Introduction

In the past decade, the research on ZnO materials has received sustained and high attention owing to its unique electrical properties, which include a wide band gap (3.37 eV), high excitation binding energy (60 meV), and large piezoelectric coefficient at room temperature [1,2,3,4]. ZnO has been demonstrated as the most promising application for electric devices. Among these ZnO nanostructures, ZnO nanorods (NRs) are the most important nanostructures. Its high surface-to-volume ratio [5], easy to fabricate [6,7,8], and one-dimensional carrier transport make it suitable for developing high performance nanoelectronic and optoelectronic devices [9,10,11], such as piezoelectric nanogenerators [12], solar cells [13,14], light emitting diodes (LED) [15], Schottky diodes [16], field effect transistors (FET) [17,18], ultraviolet sensors [19,20], and spintronics devices [21].

Moreover, the movement of electrons in a nanorod is confined to two directions, thus allowing the electron motion in one direction, leading to quantum effect. Therefore, a nanorod also shows dramatic quantum size effect, which results in large variation in electrical, chemical, and mechanical properties of the nanorod as compared to its bulk materials. Quantum effects are closely related to the axis orientation, diameter, and length of the material. The length dependence of transport properties is also widely discussed [22,23,24]. Zhou et al. [25] studied the quantum length dependence of conductance in oligomers and declared that Ohm’s law is not valid for the molecular conductance anymore. Lang et al. [26] performed first principle calculations of transport properties of carbon atom wires and found a conductance oscillation with respect to the number of carbon atoms. Garcia and Lambert [27] studied molecular wire with density functional theory (DFT) combined with the non-equilibrium Green’s function (NEGF) calculation method, and reported that the conductance is almost independent of molecular length and exhibits negative differential resistance. Hu et al. [28] reveal length dependences of the rectification ratio in organic co-oligomer. The length dependence of characteristics shows us wonderful physical phenomena, such as conductance oscillation, negative differential resistance, rectification, and so on. We are curious about the length-dependent electronic transport properties of the ZnO nanorod. Furthermore, the physical properties of a device based on the ZnO nanorod are closely related to the electron transport of the ZnO nanorod. Understanding the length-dependent transport properties of the ZnO nanorod is of great significance for the preparation of electronic devices based on the ZnO nanorod.

In this paper, we concentrated on the transport behaviors of Au-ZnONR-Au nanodevice by the DFT combined with the NEFG method. We investigate the electronic and transport properties of the triangle ZnO nanorod, which is rigidly coupled between two semi-infinite linear Au atoms. Similarly, the two-probe systems have been successfully used for the description of nanotubes [29]. The current–voltage (I–V) curve of the Au-ZnONR-Au device under the bias voltages from −1.50 V to 1.50 V is obtained. The obvious metal-like behavior and rectification characteristics are observed and analyzed.

## 2. Structure Model and Theoretical Method

Here we present an Au-ZnONR-Au nanodevice model, as shown in Figure 1. The system we studied was a typical two-probe open system, which was divided into three parts: left electrode, scattering region, and right electrode. Our central region consisted of the triangle ZnO (0001) nanorod and an electrode extension that ensured a smooth transition between the potential of the central region and electrodes. The length of the ZnO nanorod was labeled by the number of unit cells, and the length and diameter of the unit cells were 5.21 Å and 7.70 Å, respectively. The Au-ZnONR-Au nanodevice with n unit cells was named as *n*-ZnONR. In Figure 1, the left electrode directly couples the zinc atoms at the terminal of the nanorod form Au-Zn contact and the right electrode directly couples the oxygen atoms at the terminal of the nanorod form Au-O contact. The distance between the nanorod and the electrode was fixed to be a constant for all the *n*-ZnONR devices. The coupling surface distances of Au-Zn and Au-O were set to be 2.4 Å and 1.9 Å, respectively. The values were obtained from total energy optimization, and the binding energy between the ZnO nanorod and Au surface could be defined as:(1)$$E_{\mathrm{binding}}=E_{\mathrm{total}}\left[\mathrm{ZnONR}+\mathrm{Au}\right]-E_{\mathrm{total}}\left[\mathrm{ZnONR}\right]-E_{\mathrm{total}}\left[\mathrm{Au}\right]$$

The coupling surface distances of Au-Zn and Au-O were varied from 1.5 Å to 3.5 Å with a minimal step of 0.2 Å, respectively. Then, according to the lowest energy principle, it could determine the value of the coupling surface distance. This method is widely used to fix the electrode contact distance of one-dimensional nanodevices, including boron nitride nanotubes [30], ZnO nanotubes [31], and phenylene vinylene oligomere molecule systems [32].

All our calculations were carried out using the Virtual Nanolab Atomistix ToolKit (VNL-ATK, 2015.1, Synopsys, Inc., Mountain View, CA, USA) calculation package [33], which is based on the basis of the first-principles DFT combined with the NEGF formalism. Within the NEGF formalism, the current that passes through the central region at a finite bias voltage can be computed using the Landauer–Büttiker formula [34,35,36]:(2)$$I(V)=\frac{2e}{h}\int_{\mu_{L}}^{\mu_{R}}dE\left[f_{L}\left(E-V_{b}\right)-f_{R}\left(E-V_{b}\right)\right]T\left(E,V_{b}\right)$$
where *μ_L_* and *μ_R_* are the electrochemical potentials of the two electrodes, and *T*(*E*, *V_b_*) is the transmission coefficient at energy *E* and bias voltage *V_b_*. The transmission coefficient determines the probability of electrons transferring between the two semi-infinite electrodes. The transmission spectrum is calculated by the standard equation:(3)$$T\left(E,\;V_{b}\right){\;=\;T}_{r}{[\Gamma }_{L}(E,\;V_{b})\;G(E,\;V_{b}{)\Gamma }_{R}G^{†}\left(E,V_{b}\right)]$$
where *G* is the retarded Green’s function of the contact region, Γ*_L/R_* is the coupling matrix.

The equilibrium conductance (*G*) of the two-probe device was evaluated by the transmission coefficients [37].
(4)$$G=\frac{2e^{2}}{h}T\;\left(E_{f\;},V_{b}=0\;V\right)$$

In our calculations, the wave functions of all the atoms were expanded by double-zeta polarized (DZP) basis set. The electrostatic potentials were computed on a real space grid with a mesh cutoff energy of 75 hartree. The exchange correlation potential was approximated by the Perdew–Zunger parameterization of local density approximation (LDA). The 1 × 1 × 100 *k*-point was used in the x, y, and z direction, respectively. Note that the z direction is the direction of electron transport. The temperature of the electrodes was set to 300 K. A vacuum region of at least 10 Å was added to void the interaction of adjacent nanorods.

## 3. Results and Discussions

### 3.1. Transport Properties under Zero Bias Voltage

We firstly discuss the transport properties of the length-dependent ZnO nanorod at equilibrium. Taking the *n*-ZnONR (*n* = 4, 6, 8, 10) device as an example, we calculated the transmission spectrum of the device with different channel lengths, as shown in Figure 2. It can be observed that there were many transmission peaks in the energy range, which means multiple intrinsic transmission channels existed in this energy region. However, there was a different phenomenon near the Fermi level; the transmission spectrum near the Fermi level was not equal to zero in Figure 2a,b and was zero in Figure 2c,d. To explain this phenomenon, we consider that the transmission value near the Fermi level may be formed by direct tunneling from one electrode to the other. So we removed the ZnO nanorod from the two-probe system model and calculated the direct tunneling between the two electrodes at varying distances. When the distance between the two electrodes was equal to the length of one unit cell, the equilibrium conductance was 0.0324 G_0_ (G_0_ = 2e^2^/*h*). For two unit cells, the value of the transmission coefficient was zero in the range from −4 V to 4 V. So the direct tunneling could be ignored for the *n*-ZnONR device (*n* ≥ 2).

Then, we considered the influence of the interface between the Au electrode and the ZnO nanorod. Figure 3 shows the density of states (DOS) of the scattering regions, including the ZnO nanorod and 96 Au atoms that were evenly distributed at both ends of the nanorod, and the projected density of states (PDOS) of the ZnO nanorod, including only O and Zn atoms for *n*-ZnONR (*n* = 4, 6, 8, 10) devices at equilibrium. It can be observed that the shapes of the DOS of the scattering regions were similar to the PDOS of the ZnO nanorod. However, the DOS of the scattering region near Fermi level was higher than the PDOS of the ZnO nanorod, which means that the transmission spectrum near the Fermi level of the device was mainly contributed to by the interface state rather than the central ZnO nanorod. For the 4-ZnONR device, the interface states of the two terminals of the ZnO nanorod will overlap and form a transmission channel near Fermi level. Electrons can travel from the left electrode to the right electrode along the transmission channel. So the 4-ZnONR device indicates metal-like behavior near Fermi level. For the 10-ZnONR device, the scattering region had a longer nanorod. Although it had the same interface state as the 4-ZnONR device, the overlapping of the interface state decreased rapidly with the increase of the ZnO nanorod length. The effect of interface states can be ignored in the *n*-ZnONR (*n* = 8, 10) devices.

In order to further understand properties of electronic transport, the transmission eigenstates near the Fermi level were calculated, as shown in Figure 4. The transmission eigenstate corresponds to the scattering state from the left electrode to the right electrode. In Figure 4a, the eigenstates have larger amplitudes on the left side. The eigenstates corresponding to the higher transmission eigenvalue also had larger amplitude on the right side of the scattering region, which indicates that the incident state on the left side had a greater transmission probability through the intermediate region into the right electrode. The transmission eigenstates on the left side of the scattering region include the incident state and the reflective state, and the two portions have a certain superimposed interference effect. The amplitude of the eigenstate on the leftmost Zn atoms, in Figure 4a, was small due to the destructive interference of the incident state and the reflected state. In Figure 4b, the amplitude of the eigenstates almost disappearred to zero in the middle to the right of the scattering region, that is, only the scattering eigenstate was not transmitted. So the results show that the transmission eigenstates gradually changed from delocalized, throughout the channel region, to localized, at the one electrode, when the channel length increased, which indicates the carriers transport is inhibited by the increases of the channel length.

### 3.2. Transport Properties under Bias Voltage

The transmission spectrum is not sufficient to describe the transport properties of the device under zero bias voltage. It is necessary to investigate the current–voltage (I–V) curve, which can intuitively describe the electron transport properties under finite bias voltage. For a further insight into the electronic transport properties of the *n*-ZnONR devices, we presented the I–V curve for *n*-ZnONR (*n* = 4, 6, 8, 10) devices, as shown in Figure 5. The *n*-ZnONR devices showed a strong nonlinearity and reflected the Schottky-type electronic structure. The I–V curve was highly symmetric within a bias range of −1.5 V to 1.5 V. At the same bias voltage, the current value decreased with the increase of the length of the ZnO nanorod. For the 4-ZnONR device, it displayed linear features and exhibited a metal-like behavior in the low bias range (−0.75 V, 0.75 V). Comparing the I–V curves of 4-ZnONR with *n*-ZnONR *(n* = 6, 8, 10), we also found that the current of *n*-ZnONR (*n* = 6, 8, 10) increased slowly in the beginning and then increased exponentially, which is similar to a Schottky diode. 6-ZnONR had the best rectification characteristics in *n*-ZnONR (*n* = 6, 8, 10). However, there were two orders of magnitude difference compared with experiment results [38]. We think that the significant differences mainly come from two aspects. First, their size is quite different. The diameter of the nanorod in the experiment was 130 nm, while the theoretical calculation was only 0.77 nm. Second, there is quantum size effect, Li et al. [39] reported that there is a significant quantum size effect when the diameter of the ZnO nanowire is less than 2.8 nm.

In order to better understand the rectification characteristics caused by dependence length for the ZnO nanorod device, it is necessary to investigate the change of the transmission spectra under the applied bias. Transmission spectra of *n*-ZnONR (*n* = 4, 6) devices at different bias voltages are shown in Figure 6. According to Equation (2), the current was contributed from the energy integral in the bias window region. Here, the Fermi level that is the average of *μ_L_* and *μ_R_* was set as zero. Firstly, in the case of the 4-ZnONR device, a small value of transmission peak was observed at 0.08 eV in the bias window from −0.2 eV to 0.2 eV, in Figure 6a, which resulted in current. With the increase of bias voltage, more transmission peaks were located on the bias window, as shown in Figure 6b,c, hence the current increased linearly. In Figure 6d,e, the bias voltage was greater than 1.6 eV, which was mainly because of the fact that the large value of transmission peaks appearred in the bias window, which resulted in the marked increase of the current. In Figure 6f,g, the bias voltage is 0.4 V and 0.8 V, respectively. There is no transmission spectrum in the bias window for the 6-ZnONR device. In Figure 6h, the transmission spectrum was located on the bias window and generated a current. There were many transmission peaks above the Fermi level at 2.0 V and a transmission peak was also observed below the Fermi level, as shown Figure 6k, which was an exponential increase in current.

The electron transport properties can also be analyzed using the spatial distribution of the frontier molecular orbital. The orbit corresponding to the first transmission peak below the Fermi level is the highest occupied molecular orbital (HOMO). Similarly, the orbit corresponding to the first transmission peak above the Fermi level is the lowest unoccupied molecular orbital (LUMO). The nature of the HOMO and LUMO of the molecular-projected self-consistent Hamiltonian (MPSH) provides some phenomenon of the rectification characteristics. The eigenstate of MPSH determines the transmission peak, which are linked to the poles of the Green’s function [40]. These are relevant to the transmission spectra within the bias window [41]. In general, the molecular energy orbit levels determine the position of the transmission peak, and the height of the transmission peak is correlated with the delocalization of the molecular orbitals. As shown in Figure 7, it clearly shows that the spatial distribution of MPSH states corresponding to LUMO + 1, LUMO, HOMO, and HOMO − 1 of the 6-ZnONR device was at 0.8 V, and 1.6 V. At the bias voltage of 0.8 V, the orbitals were all localized near the right electrode, which was difficult to form a transmissive channel, resulting in a zero circuit. However, at the bias voltage of 1.6 V, the orbitals were fully delocalized, which was easy to form a transmission channel, resulting in a large circuit.

## 4. Conclusions

In conclusion, we computed transmission spectra, DOS, and I–V curve of *n*-ZNONR devices by using the DFT method coupled with the NEGF. It was found that the contact interfaces between Au electrodes and the ZnO nanorod play essential roles in the current–voltage behaviors. For the 4-ZnONR device, the metal–nanorod interface states overlap to form a transmission channel, and results in the linear behavior of the electrical transport at the low bias voltage. As the ZnO nanorod length is increased, the central scattering region determines the transport characteristics of the device and exhibits excellent rectification characteristics. The rectification characteristics were analyzed using transmission spectra at different bias voltages and MPSH. Our results would have theoretical guidance for the preparation of field effect transistors based on the ZnO nanorod.

## Figures and Tables

**Figure 1 micromachines-10-00026-f001:**
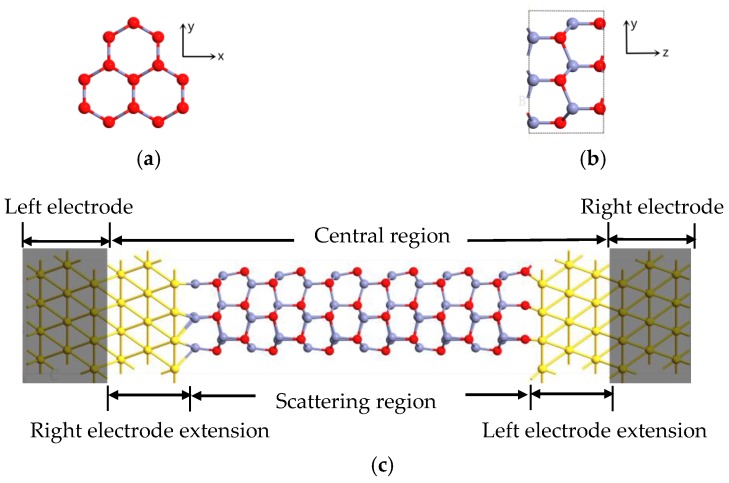
Schematic of the two-probe device structure model. (**a**) The *x*–*y* plane of the unit cell. (**b**) The *z*–*y* plane of the unit cell. (**c**) The shadow areas at the ends of the model indicate the left and right electrodes, the channel length is indicated by the number of unit cells of the ZnO (Zn and O atoms are denoted with gray and red, respectively).

**Figure 2 micromachines-10-00026-f002:**
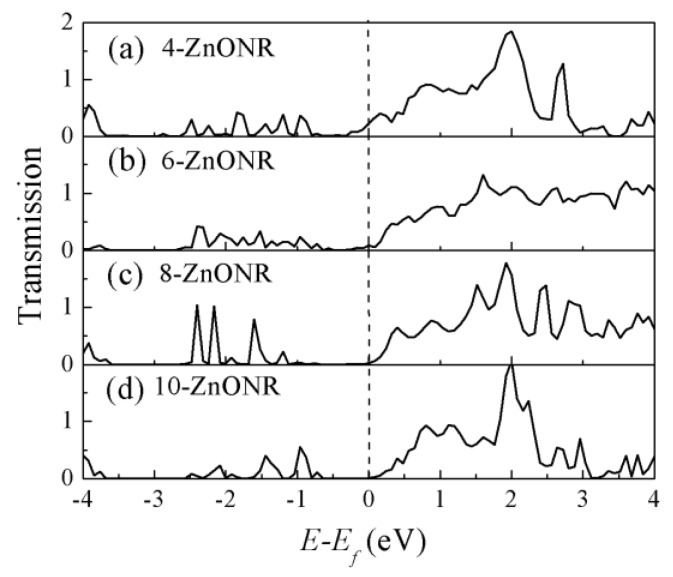
Transmission spectrum of the *n*-ZnONR ((**a**): *n* = 4, (**b**): *n* = 6, (**c**): *n* = 8, (**d**): *n* = 10) device at equilibrium, and the zero of the energy is set to Fermi level.

**Figure 3 micromachines-10-00026-f003:**
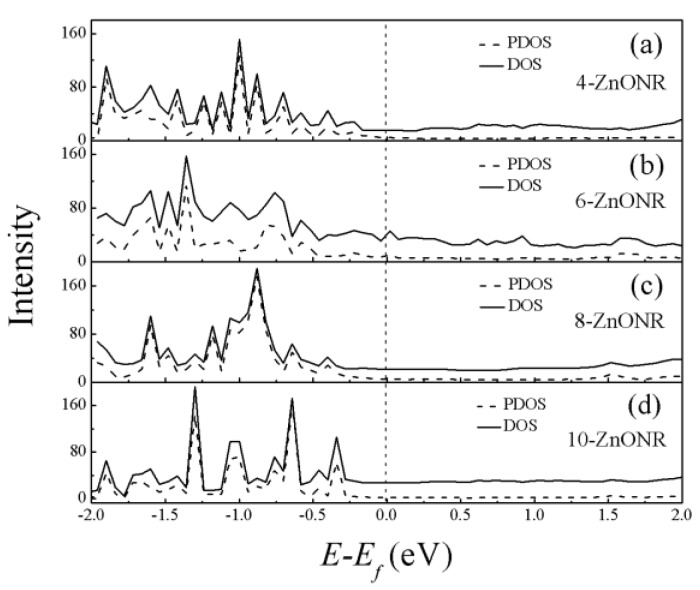
Density of states (DOS) of the scattering regions and projected density of states (PDOS) of the *n*-ZnONR ((**a**) 4-ZnONR, (**b**) 6-ZnONR, (**c**): 8-ZnONR, (**d**) 10-ZnONR) for different lengths at equilibrium, and the zero of the energy is set to Fermi level.

**Figure 4 micromachines-10-00026-f004:**
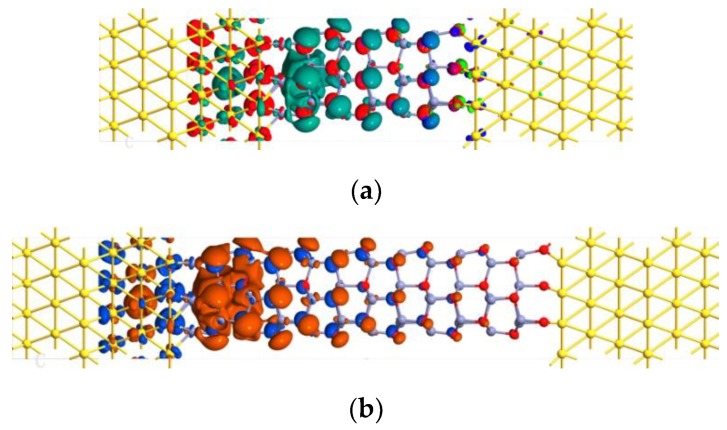
The transmission eigenstates of the *n*-ZnONR device at $E_{f}$ with an isovalue of 0.2. (**a**) (*n* = 4), (**b**) (*n* = 10), and the zero of the energy is set to Fermi level.

**Figure 5 micromachines-10-00026-f005:**
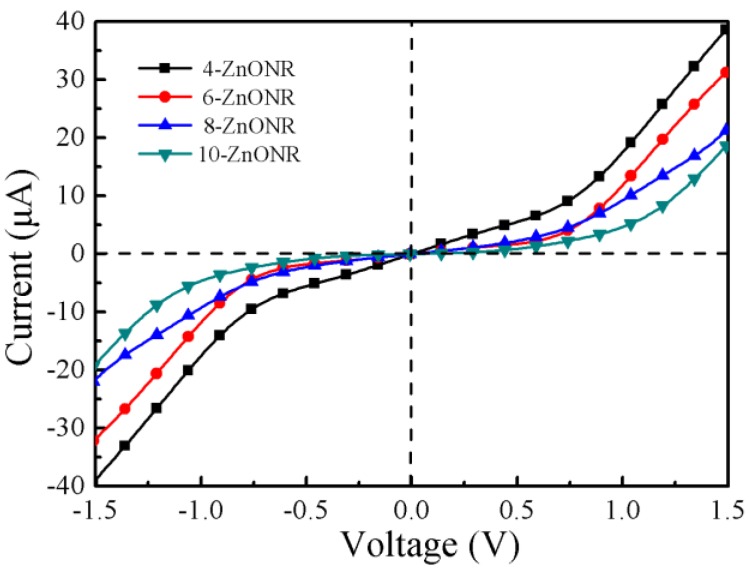
Current–voltage (I–V) curves of the *n*-ZnONR (*n* = 4, 6, 8, 10) devices.

**Figure 6 micromachines-10-00026-f006:**
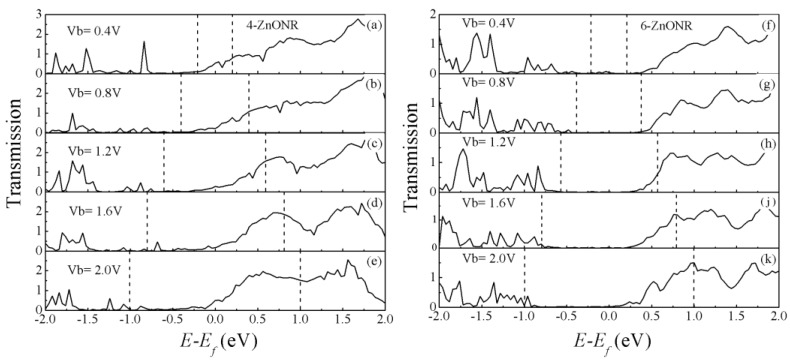
Transmission spectra of 4-ZnONR (**a**–**e**) and 6-ZnONR (**f**–**k**) devices under bias voltage. The dotted lines represent bias windows and the zero of the energy is set to Fermi level.

**Figure 7 micromachines-10-00026-f007:**
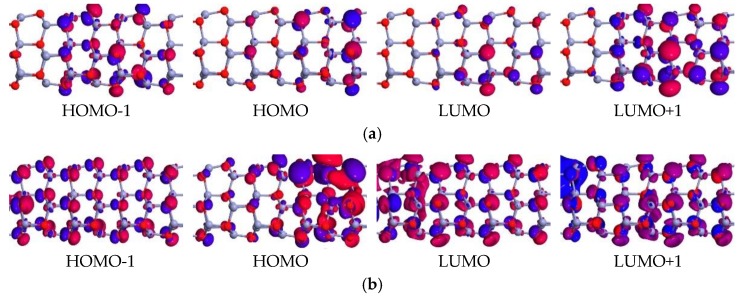
Spatial distribution of molecular-projected self-consistent Hamiltonian (MPSH) states of 6-ZnONR, (**a**) 0.8 V, (**b**) 1.6 V, and the zero of the energy is set to Fermi level.

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
