# Peer review of "Length-Dependent Electronic Transport Properties of the ZnO Nanorod"

_micromachines, 2018, doi:10.3390/mi10010026_

Round 1

Reviewer 1 Report

The work presents a nice complete theoretical story of the length-dependent electronic transport properties of ZnO nanowires. However, there are a few key points that the authors should address before publication. 

- In general, it would be nice for the authors to highlight key results in the introduction rather than simply list studies. For example, line 42, what is the relationship between piezoelectric coefficient and diameter? This will help the reader. 

-Line 44 - mechanisms instead of mechanics?

-Line 46 - it is unclear if the authors claim that there has only been theoretical studies into length-dependent transport or few studies. There have been a few experimental studies into the size effect on ZnO nanowires (eg 10.1002/chem.201204429, http://www.ee.bgu.ac.il/~shalish/papers/prb8.pdf). It is important that the authors note these studies and discuss whether their findings compare or not, 

- The description of the material measured is not clear. What is meant by 3-buffer layer electrodes? Is it simply one atomic layer of gold between the NW and the electrodes? This needs to be made more clear and can help by adding this information to the schematic diagram in Figure 1. 

-Figure 1 needs to label clearly that the unit cells are depicted by the dashed boxes. 

-It is also not mentioned what the length of a single unit cell is. Please note this to compare to experiments. 

-Please state the values used for the electrochemical potentials of the electrodes. 

- Line 96-97 - It is unclear where the vacuum is added, could you explicitly explain this in the text. Also this is the first time that adjacent nanowires are mentioned. Does this model apply boundary conditions to assume an infinite array of nanowires? Please make this clear. 

- For all figures, it is unclear the position of the Fermi Level. I appreciate that it has been set to zero for ease, but could you please make this clear on the figures and in the text. Maybe the axis should be E-Ef on the figures or the Fermi energy clearly labelled on the diagrams. 

-Line 112 - it is mentioned that the tunnelling from electrodes does not contribute for 4 unit cells. What is the number of unit cells that this effect becomes significant? Could you state in the text the length of nanowire that this must be considered for. 

-Line 118 - how many Au atoms? How is this determined or set?

-For Figure 3, the data seems to be truncated. Please show the full intensity amplitude rather than cut the y-axes on these figures to cut the data. There are also no labels for the no of unit cells on the diagram. Is a) for 4 unit cells and d) 10? Please label clearly. 

-Line 136 - amplitude is used here ? But the phase is plotted? Please clearly explain what you mean by the terminology. It would also be helpful to label the scattering region in Figure 4 on the diagram. 

-Line 155 - I am not sure what you mean by effectively described 

-The IV curves are extremely interesting, it would be nice for the authors to correlate their results back to the explanation given due to localisation of the eigenstates to explicitly explain how the eigenstate determines the electron transport. 

-Figure 6 again needs clear labelling. The caption appears incorrect, as I assume that the left column is for 4 unit cells and the right is for 6 unit cells. This needs to be labelled in caption and figure and the Fermi level also stated that it is set to zero on the figure. 

Author Response

Thank you for your letter and the reviewer’s comments concerning our manuscript entitled “Length-dependent electronic transport properties of ZnO nanorod” (Manuscript ID: micromachines-403552). Those comments are all valuable and very helpful for revising and improving our paper, as well as the important guiding significance to our researches. We have made changes according to the editor’s and reviewer’s comments and suggestions in this revised version. The revised portions are marked in red in the paper. Below we will answer in detail to the points raised by the reviewers.

Response to Reviewer

[Comment]:

The work presents a nice complete theoretical story of the length-dependent electronic transport properties of ZnO nanowires. However, there are a few key points that the authors should address before publication. 

1. In general, it would be nice for the authors to highlight key results in the introduction rather than simply list studies. For example, line 42, what is the relationship between piezoelectric coefficient and diameter? This will help the reader. 

[Answer]: Thanks for your comments. We have revised our work again and the revised portions are marked in red in the paper.

We recognize that it is not good to simply list the results of the research, so we have made significant changes to the introduction.

The length dependence of transport properties is also widely discussed [22-24]. Zhou et al [25] studied the quantum length dependence of conductance in oligomers and declared that ohm’s law is not valid for the molecular conductance anymore. Lang et al [26] performed first principles calculations of transport properties of carbon atom wires and found a conductance oscillation with respect to the number of carbon atom. Garcia and Lambert [27] studied molecular wire with DFT combined with NEGF calculation method and report that the conductance is almost independent of molecular length and exhibit negative differential resistance. Hu et al [28] reveal length dependences of the rectification ratio in organic co-oligomer. The length dependence of characteristics shows us wonderful physical phenomena, such as conductance oscillation, negative differential resistance, rectification and so on. We are curious about length dependence electronic transport properties of ZnO nanorod. Furthermore, the physical properties of device based on ZnO nanorod are closely related to the electron transport of the ZnO nanorod. Understanding the length dependence transport properties of ZnO nanorod is of great significance for the preparation of electronic devices based on ZnO nanorod.

2. Line 44 - mechanisms instead of mechanics?

[Answer]: Thanks for your comments. We have revised our work again.

We think that the mechanisms instead of mechanics are very reasonable, but we have made major changes to the introduction, and the mechanisms do not exist in the text.

3. Line 46 - it is unclear if the authors claim that there has only been theoretical studies into length-dependent transport or few studies. There have been a few experimental studies into the size effect on ZnO nanowires (eg 10.1002/chem.201204429, http://ww.ee.bgu.ac.il/~shalish/papers/prb8.pdf). It is important that the authors note these studies and discuss whether their findings compare or not, 

[Answer]: Thanks for your comments. We have revised our work again and the revised portions are marked in red in the paper.

I am very grateful to provide me with relevant literature. I also consulted the literature and compared the experimental data with the calculated data.

There are two orders of magnitude difference compared with experiment result [37]. We think that the significant differences mainly come from two aspects. First, their size is quite different. The diameter of the nanorods in the experiment is 130 nm, while the theoretical calculation is only 0.77 nm. Second, there is quantum size effect, Li et al [38] reported that there is a significant quantum size effect when the diameter of the ZnO nanowire is less than 2.8 nm.

4. The description of the material measured is not clear. What is meant by 3-buffer layer electrodes? Is it simply one atomic layer of gold between the NW and the electrodes? This needs to be made more clear and can help by adding this information to the schematic diagram in Figure 1. 

[Answer]: Thanks for your comments. We have revised our work again.

In order to make the statement more clear. We replaced the 3-buffer layer electrode with the electrode extension and detailed it in Figure 1.

5. Figure 1 needs to label clearly that the unit cells are depicted by the dashed boxes. 

[Answer]: Thanks for your comments. We have revised our work again.

To be more intuitive, we add the x-y plan and z-y plan of the unit cell in Figure 1.

6. It is also not mentioned what the length of a single unit cell is. Please note this to compare to experiments. 

[Answer]: Thanks for your comments. We have revised our work again and the revised portions are marked in red in the paper.

We have added a single unit cell size parameter in the text.

The length of ZnO nanorod is labeled by the number of unit cell, and the length and diameter of unit cell are 5.21 Å and 7.70 Å, respectively.

7. Please state the values used for the electrochemical potentials of the electrodes. 

[Answer]: Thanks for your comments. We have revised our work again.

In the calculation, the voltage is directly applied to the electrode. therefore, the bias voltage in the text is the electrochemical potentials of the electrodes.

8. Line 96-97 - It is unclear where the vacuum is added, could you explicitly explain this in the text. Also this is the first time that adjacent nanowires are mentioned. Does this model apply boundary conditions to assume an infinite array of nanowires? Please make this clear. 

[Answer]: Thanks for your comments. We have revised our work again and the revised portions are marked in red in the paper.

In the theoretical calculation by VNL-ATK, if you do not add sufficient vacuum boundary conditions, the system will default to a periodic structure. For example, if the vacuum region is not set in the paper, the model is an infinite ZnO array of nanorods. In this paper, we study the transport properties of individual ZnO nanorod. It is necessary to add sufficient vacuum boundary conditions to destroy its periodicity and prevent the formation of ZnO array.

A vacuum region of at least 10 Å is added to void the interaction of adjacent nanorods.

9. For all figures, it is unclear the position of the Fermi Level. I appreciate that it has been set to zero for ease, but could you please make this clear on the figures and in the text. Maybe the axis should be E-Ef on the figures or the Fermi energy clearly labelled on the diagrams. 

[Answer]: Thanks for your comments. We have revised our work again.

We set the Fermi level to zero in the calculations and figures, but we didn't make it clear in the text. Now, we make this clear on the figures and in the text.

10. Line 112 - it is mentioned that the tunnelling from electrodes does not contribute for 4 unit cells. What is the number of unit cells that this effect becomes significant? Could you state in the text the length of nanowire that this must be considered for. 

[Answer]: Thanks for your comments. We have revised our work again and the revised portions are marked in red in the paper.

We supplemented the calculation of electrode tunneling that the distance between the two electrodes is equal to the length of one unit cell and two unit cell.

When the distance between the two electrodes is equal to the length of one unit cell, the equilibrium conductance is 0.0324 G0 (G0=2e2/h). For two unit cells, the value of transmission coefficient is zero in the range from -4 V to 4 V. So the direct tunneling could be ignored for n-ZnONW device (n2) .

11. Line 118 - how many Au atoms? How is this determined or set?

[Answer]: Thanks for your comments. We have revised our work again and the revised portions are marked in red in the paper.

When optimizing the distance between the Au-Zn and Au-O coupling faces, we found that due to the lattice mismatch, the gold atoms in the outermost layer are displaced, while the gold atoms in the outer layer have almost no displacement. Therefore, we have chosen all the gold atoms of the electrode extension to fully guarantee the calculation requirements.

96 Au atoms that is evenly distributed at both ends of the nanorod.

12. For Figure 3, the data seems to be truncated. Please show the full intensity amplitude rather than cut the y-axes on these figures to cut the data. There are also no labels for the no of unit cells on the diagram. Is a) for 4 unit cells and d) 10? Please label clearly. 

[Answer]: Thanks for your comments. We have revised our work again.

        We have expanded the coordinate values of the y-axis and all the data is shown in the figure. The labels are also more complete.

13. Line 136 - amplitude is used here? But the phase is plotted? Please clearly explain what you mean by the terminology. It would also be helpful to label the scattering region in Figure 4 on the diagram. 

[Answer]: Thanks for your comments. We have revised our work again.

In Figure 4, the transmission eigenstate contains amplitude and phase information. The amplitude information is represented by the isosurface, and the phase information is distinguished by color. The phase relationship is not considered in the text, we removed the phase label.

14. Line 155 - I am not sure what you mean by effectively described.

[Answer]: Thanks for your comments. We have revised our work again and the revised portions are marked in red in the paper.

Here we have problems that are unclear. We have revised it.

The transmission spectrum is not sufficient to describe the transport properties of the device under zero bias voltage. It is necessary to investigate the current-voltage (I-V) curve which can intuitively describe the electron transport properties under finite bias voltage. 

15. The IV curves are extremely interesting, it would be nice for the authors to correlate their results back to the explanation given due to localisation of the eigenstates to explicitly explain how the eigenstate determines the electron transport. 

[Answer]: Thanks for your comments. We have revised our work again.

Thank you for agreeing with the I-V curve we calculated.  According to formula 2, the current value is obtained by integrating the transmission spectrum in the bias window. The peak of transmission spectrum in bias window has important influence on the current value. The eigenstate determines the position of the transmission peak. Therefore, we use transmission spectrum and eigenstates to analyze the I-V curve in detail.

16. Figure 6 again needs clear labelling. The caption appears incorrect, as I assume that the left column is for 4 unit cells and the right is for 6 unit cells. This needs to be labelled in caption and figure and the Fermi level also stated that it is set to zero on the figure. 

[Answer]: Thanks for your comments. We have revised our work again.

We added labell in the caption and fugure, and set the Fermi level to zero.

Transmission spectra of 4-ZnONW (a)-(e) and 6-ZnONW (f)-(k) device under bias voltage. The dotted lines represent bias windows and the zero of energy is set to Fermi level.

We tried our best to improve the manuscript and made some changes in the manuscript. These changes will not influence the content and framework of the paper.

We appreciate for Editors/Reviewers’ warm work earnestly, and hope that the correction will meet with approval.

Once again, thanks very much for your comments and suggestions.

Reviewer 2 Report

The research idea in presented in this paper is sound and interesting. However, there are some revision should be done to improve the quality of the paper.

1.                Line 52, "triangle ZnO nanowires", why is triangle instead of hexagonal? What is the diameter of the nanowire?

2.                The introduction is very general. For example, in the second paragraph, the authors focus on the quantum size effect and list quite some publication but did not go into detail what the conclusions are. It should be improved.

3.                In the paper, sometimes it was written DFT while sometimes it was written “density functional theory”, so are “NEGF” and “nonequilibrium Green’s function method” please unify them.

4.                Line 71 and 75, “the coupling surface distances” of Au-Zn and Au-O, what is the physical meaning of this distance? Is it related to the structure mismatch?

5.                Figure 5 is very different with the experimental result Li et al reported (ref 27). The reason should be discussed.

6.                This paper is about “length-dependent” electronic transport properties of ZnO, but the authors only calculated 4, 6, 8 and 10 unitcells, why not expend the result to 20 or more, would be very interesting and closer to experimental model.

7.                ZnO is known to be an intrinsic n-type semiconductor. However, it was not mentioned in the text and the I-V curve do nor represent the rectifying behavior. Is there any special reason?

8.                The conclusion is very general. Line 210 and 211 “our results would have theoretical guidance for the experiments and applications in ZnO nanowire electronic devices”. What kind of ZnO electronic device? It should be more specific.

9.                nanowire/NW is used in the title and all over the text. which is not appropriate consider the aspect ratio of the structures which are studied in this paper. Please change them to “nanorod”.

Author Response

Thank you for your letter and the reviewer’s comments concerning our manuscript entitled “Length-dependent electronic transport properties of ZnO nanorod” (Manuscript ID: micromachines-403552). Those comments are all valuable and very helpful for revising and improving our paper, as well as the important guiding significance to our researches. We have made changes according to the editor’s and reviewer’s comments and suggestions in this revised version. The revised portions are marked in red in the paper. Below we will answer in detail to the points raised by the reviewers.

Response to Reviewer

[Comment]:

The research idea in presented in this paper is sound and interesting. However, there are some revision should be done to improve the quality of the paper.

1. Line 52, "triangle ZnO nanowires", why is triangle instead of hexagonal? What is the diameter of the nanowire?

[Answer]: Thanks for your comments. We have revised our work again and the revised portions are marked in red in the paper.

ZnO nanorod is a hexagonal structure that forms a triangular morphologies. This morphologies of ZnO nanowire has been prepared in the laboratory(eg doi:10.1016/j.cplett.2006.04.013). Its diameter is 0.77 nm.  We added the parameters of the unit cell in the text.

The length of ZnO nanorod is labeled by the number of unit cell, and the length and diameter of unit cell are 5.21 Å and 7.70 Å, respectively.

2. The introduction is very general. For example, in the second paragraph, the authors focus on the quantum size effect and list quite some publication but did not go into detail what the conclusions are. It should be improved.

[Answer]: Thanks for your comments. We have revised our work again and the revised portions are marked in red in the paper.

We have carefully revised the introduction.

The length dependence of transport properties is also widely discussed [22-24]. Zhou et al [25] studied the quantum length dependence of conductance in oligomers and declared that ohm’s law is not valid for the molecular conductance anymore. Lang et al [26] performed first principles calculations of transport properties of carbon atom wires and found a conductance oscillation with respect to the number of carbon atom. Garcia and Lambert [27] studied molecular wire with DFT combined with NEGF calculation method and report that the conductance is almost independent of molecular length and exhibit negative differential resistance. Hu et al [28] reveal length dependences of the rectification ratio in organic co-oligomer. The length dependence of characteristics shows us wonderful physical phenomena, such as conductance oscillation, negative differential resistance, rectification and so on. We are curious about length dependence electronic transport properties of ZnO nanorod. Furthermore, the physical properties of device based on ZnO nanorod are closely related to the electron transport of the ZnO nanorod. Understanding the length dependence transport properties of ZnO nanorod is of great significance for the preparation of electronic devices based on ZnO nanorod.

3. In the paper, sometimes it was written DFT while sometimes it was written “density functional theory”, so are “NEGF” and ““density functional theory” please unify them.

[Answer]: Thanks for your comments. We have revised our work again and the revised portions are marked in red in the paper.

We unified the format.

In this paper, we concentrated on the transport behaviors of Au-ZnONW-Au nanodevice by the DFT combined with NEFG method.

4. Line 71 and 75, “the coupling surface distances” of Au-Zn and Au-O, what is the physical meaning of this distance? Is it related to the structure mismatch?

[Answer]: Thanks for your comments. We have revised our work again and the revised portions are marked in red in the paper.

There is the structure mismatch between Au and ZnO during the coupling process. According to the principle of minimum energy, we set different coupling distances for geometric optimization. Quantum ATK now scans all possible repetitions and rotations of two surface in order to find a common supercell with minimal strain. At this time, the system has the lowest energy and is in a stable state. This method is widely used in first-principles calculations.

5. Figure 5 is very different with the experimental result Li et al reported (ref 27). The reason should be discussed.

[Answer]: Thanks for your comments. We have revised our work again and the revised portions are marked in red in the paper.

After the nanowires are modified to nanorods, I consulted the literature and discussed the difference between experiment and calculation.

There are two orders of magnitude difference compared with experiment result [37]. We think that the significant differences mainly come from two aspects. First, their size is quite different. The diameter of the nanorods in the experiment is 130 nm, while the theoretical calculation is only 0.77 nm. Second, there is quantum size effect, Li et al [38] reported that there is a significant quantum size effect when the diameter of the ZnO nanowire is less than 2.8 nm.

6. This paper is about “length-dependent” electronic transport properties of ZnO, but the authors only calculated 4, 6, 8 and 10 unit cells, why not expend the result to 20 or more, would be very interesting and closer to experimental model.

[Answer]: Thanks for your comments. We have revised our work again.

The experimental model is accurately simulated, which is the direction and goal of theoretical calculations. At present, The typical two probe open system is used to study the transport characteristics, as shown in Figure 1. The left and right electrodes are semi-infinite periodic structures, and the scattering region is a non-periodic structure, which is also a key part of the device. If the size of computational model is close to the size of experimental model, there is a huge amount of computation that can’t be done. Now, our servers can only complete the calculation of 10-ZnONR devices, and we hope to expand to 20 or more by upgrading our servers in the future.

7. ZnO is known to be an intrinsic n-type semiconductor. However, it was not mentioned in the text and the I-V curve do nor represent the rectifying behavior. Is there any special reason?

[Answer]: Thanks for your comments. We have revised our work again and the revised portions are marked in red in the paper.

We all know that intrinsic ZnO materials are n-type semiconductor materials, and n-type is mainly caused by intrinsic oxygen defects. Our model is ideal for ZnO nanorods. Therefore, we don’t mention n-type semiconductors in the text. The rectification behavior represented by the I-V curve is not ideal, and we think it still has a little rectification behavior. We hope that the review will agree with our views.

8. The conclusion is very general. Line 210 and 211 “our results would have theoretical guidance for the experiments and applications in ZnO nanowire electronic devices”. What kind of ZnO electronic device? It should be more specific.

[Answer]: Thanks for your comments. We have revised our work again and the revised portions are marked in red in the paper.

Field effect transistors were fabricated using individual ZnO nanorod. Its performance mainly depends on the transport properties of ZnO nanorod. The I-V curve of individual ZnO nanorod is helpful to the preparation of field effect transistors based on ZnO nanorod.

 Our results would have theoretical guidance for the preparation of field effect transistors based on ZnO nanorod.

9. nanowire/NW is used in the title and all over the text. which is not appropriate consider the aspect ratio of the structures which are studied in this paper. Please change them to “nanorod”.

[Answer]: Thanks for your comments. We have revised our work again and the revised portions are marked in red in the paper.

According to the aspect ratio of the model structure, we also think that it is more reasonable to change nanowires to nanorods. We changed the nanowires to nanorods and changed the related references in the text.

We tried our best to improve the manuscript and made some changes in the manuscript. These changes will not influence the content and framework of the paper.

We appreciate for Editors/Reviewers’ warm work earnestly, and hope that the correction will meet with approval.

Once again, thanks very much for your comments and suggestions.
